# Evaluation of substrate composition and exogenous hormone application on vegetative propagule rooting success of essential oil hemp (*Cannabis sativa* L.)

**Sean M. Campbell**[1¤], **Steven L. Anderson**[1], **Zachary T. Brym**[2], **Brian J. Pearson**[1]*

**1** Department of Environmental Horticulture, Mid-Florida Research and Education Center, University of Florida, Institute of Food and Agricultural Sciences, Apopka, Florida, United States of America, **2** Department of Agronomy, Tropical Research and Education Center, University of Florida, Institute of Food and Agricultural Sciences, Homestead, Florida, United States of America

¤ Current address: Department of Viticulture and Enology, University of California, Davis, Davis, California, United States of America
* bpearson@ufl.edu

**Data Availability Statement:** All relevant data are within the manuscript and its S1 File.

## Abstract

To support the rapidly expanding industrial hemp industry, a commercial supply of high-quality starter plants with low genetic variability from nurseries will be key to consistent and efficient cultivation efforts. Rooting success was evaluated across four propagation medias, five rooting hormones, and eight commercially available high-cannabidiol (CBD) essential oil hemp cultivars. Cuttings were placed in a climate-controlled room and assessed for rooting success 12 days after cloning. Rooting success was determined by quantifying total root number, cumulative total root length, and total root mass. Propagation media had the greatest effect on rooting success (13–80%). Rockwool had the highest rooting success resulting in 10-fold increases in rooting traits over the next highest scoring medium (Berger BM6). Hormone applications significantly improved (15- to 18-fold) rooting success compared to no hormone application, while non-statistical differences were observed across auxin hormone concentrations and application methods. Genetic variation in rooting response was observed between cultivars with 'Cherry Wine' outperforming all other cultivars with an approximate 20% increase in rooting success over the next highest rooting cultivar, 'Wife'. Although the ideal combination was not specifically identified in this study, findings provide insight into how rooting hormone application and medium selection impact vegetative propagule rooting success of essential oil hemp.

## 1. Introduction

Hemp (*Cannabis sativa* L.) is a short day, herbaceous, commonly dioecious annual plant cultivated throughout the world for grain, fiber, and secondary metabolites. Passing of the Agriculture Improvement Act of 2018 federally commercialized production and distribution of industrial hemp [*C. sativa* having less than 0.3% delta-9 tetrahydrocannabinol (THC)] in the

**Funding:** The funders had no role in study design, data collection and analysis, decision to publish, or preparation of the manuscript. This project was made possible by financial support from Green Roads, LLC and the UF/IFAS Office of the Dean and Research. Green Point Research, Green Roads LLC, and ANO Colorado LLC donated the cultivars used in this research. Steven Anderson was funded by Roseville Farm's UF/IFAS Florida Industrial Hemp Endowment contribution.

**Competing interests:** The authors declare that funds were received from Green Roads, LLC and Roseville Farms to conduct this research and fund salaries. This does not alter our adherence to PLOS ONE policies on sharing data and materials.

United States [1]. Of the $1.4 billion in hemp-based product sales in the U.S. for 2019, $813.2 million were attributed to consumer sales of cannabidiol (CBD), a secondary metabolite, and its associated products. These values are estimated at $2.61 and $1.81 billion, respectively, by 2024 [2, 3]. Unlike grain and fiber varieties which are planted exclusively by seed, essential oil hemp is cultivated through seed and cutting propagation. Seed propagation of essential oil hemp is simple and cost-effective, but commonly lacks genetic uniformity required for commercial cultivation [4, 5]. Vegetative propagation guarantees genetic uniformity and production of female plants, an important attribute for dioecious essential oil hemp. Male plants inherently produce low levels of oil and, through sexual fertilization, can reduce oil production in female plants by as much as 75% [6]. Therefore, asexual vegetative propagation of hemp offers a reliable, cost-effective method for hemp cultivators and liner producers.

Successful rooting of vegetative cuttings can be influenced by an array of factors including plant genetics [7], cutting caliper [8, 9], humidity [10], temperature [11, 12], season [8, 9], mother stock age/health [9, 12], leaf number and tipping [13], hormone application [13], and stem wounding [7]. Currently, limited scientific literature exists with respect to optimizing vegetative propagation in *Cannabis*. Caplan et al. [13] demonstrated uncut leaf tips and application of 2000 ppm indole-3-butyric (IBA) acid gel significantly increased rooting success of *Cannabis*, while removal of leaves (3 versus 2), and location of cutting (apical versus basal) had little effect on rooting success. Campbell et al. [7] demonstrated stem wounding followed by 1000 ppm IBA significantly increased rooting success and time to rooting across three marijuana cultivars. Of the current *Cannabis* rooting studies, environmental conditions consisted of growth chamber or humidity domes which may not be applicable at commercial scale. Hormone rates, application methods, and rooting media have not been empirically studied to our knowledge and could significantly optimize propagation success.

Optimal propagation media depends upon plant species, cutting type, season, and propagation system. Rooting medium serves several functions towards rooting success, to include: (i) holding the cutting in place during rooting, (ii) providing moisture to the cutting, (iii) allowing air to the base of the cutting, and (iv) reducing light penetration to cutting base [14]. Ideal propagation medium has sufficient porosity combined with high water holding capacity while also being well drained. Optimal water retention of medium is critical to propagation success as excess water retention can block oxygen to developing roots [15]. Furthermore, propagation systems in conjunction with season may dictate the appropriate propagation media for optimized rooting success. Commonly, propagators use a combination of organic (peat, sphagnum moss, bark) and mineral components (perlite, vermiculite, polystyrene, rockwool) [16] in combination with intermittent misting, fog, or enclosed systems.

Treating cuttings with auxin increases formation of roots, reduces time to root initiation, and improves rooting uniformity [14, 17]. Valued for their extensive history of use and consistent rooting response, indole-3-butyric acid (IBA) and 1-naphthalenacetic acid (NAA) are common auxins used in commercial propagation of plants [17, 18]. Auxin is commonly applied to cuttings via liquid solution, powder formulations, or a combination of both methods [14], although liquid solutions are generally more effective than powder formulations [19, 20]. Low concentrations of IBA (1000–2000 ppm) have demonstrated improved rooting success of *Cannabis* compared to no auxin application [7, 13], indicating hemp rooting lies within the range of softwood/herbaceous to semi-hardwood auxin application rates and may tolerate auxin levels up to 5000 ppm [14]. No comparison of auxin application rates has been empirically study in *Cannabis* to our knowledge.

Where propagative information is made available, contributions are primarily sourced from *Cannabis*-based website opinions and user forums, thus there is a need for reliable, empirically derived information that has examined factors influential in rooting success of

essential oil hemp. The objectives of this investigation were to examine the influence of four media compositions and five exogenous plant rooting hormone compounds on rooting success of eight vegetatively propagated essential oil hemp cultivars. The objectives of this study were to: (i) quantify the effects of commercially available rooting mediums on rooting success of essential oil hemp, (ii) evaluate the effect of commercial auxin products of varying concentrations and application methods on rooting success of essential oil hemp cultivars, and (iii) determine if genetic variation in rooting response exists across essential oil hemp cultivars.

## 2. Materials and methods

### 2.1 Propagation media

Four soilless media were selected based on their bulk density ($D_b$) and water holding capacities as presented by the manufacturer's product specifications, which was mainly influenced by percentage peat (Fig 1). Propagation media included:

- PRO-MIX High Porosity (HP) Mycorrhizae Growing Medium (Premier Tech Horticulture; Quakertown, PA).

- Jolly Gardener Pro-Line C/20 (Oldcastle Lawn & Garden; Atlanta, GA).

- Berger BM6 All-Purpose Mix (Berger; Saint-Modeste, QC).

- Stone Wool A-OK 1.5" Starter Plugs (Grodan; Roermond, The Netherlands), hereafter referred to as Grodan rockwool will be used as reference.

### 2.2 Rooting hormone

Five rooting hormone treatments were prepared according to manufacturer recommendations and included: Hormidin 1 (H1, 1,000 ppm IBA); Hormidin 2 (H2, 3,000 ppm IBA); 1:10 Dip 'N Grow (DNG, 1,000 ppm IBA/500 ppm NAA); 1:5 Hormex (Hx, 500 ppm NAA); and an untreated control. H1 and H2 (OHP, Inc., Bluffton, SC) were applied directly in their powdered form to the freshly cut portion of the internode. The DNG (Dip 'N Grow Inc, Clackamas, OR) liquid concentrate was diluted to a 1:10 ratio with distilled water, reducing it from the packaged 10,000 ppm IBA/5,000 ppm NAA to the desired 1,000 ppm IBA/500 ppm NAA.

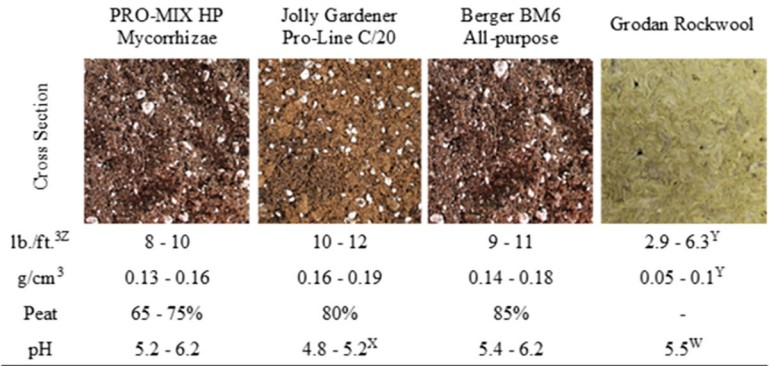

| | PRO-MIX HP Mycorrhizae | Jolly Gardener Pro-Line C/20 | Berger BM6 All-purpose | Grodan Rockwool |
|---|---|---|---|---|
| Cross Section | | | | |
| lb./ft.$^{3Z}$ | 8 – 10 | 10 – 12 | 9 – 11 | 2.9 – 6.3$^Y$ |
| g/cm$^3$ | 0.13 – 0.16 | 0.16 – 0.19 | 0.14 – 0.18 | 0.05 – 0.1$^Y$ |
| Peat | 65 – 75% | 80% | 85% | - |
| pH | 5.2 – 6.2 | 4.8 – 5.2$^X$ | 5.4 – 6.2 | 5.5$^W$ |

**Fig 1. Cross section, dry bulk density in lb./ft.$^3$ and g/cm$^3$, sphagnum peat moss content (%) and pH for the four experimental substrates.** [X]Values determined experimentally using an average of three replicates of the saturated paste extract method [21]. [W]Rockwool pH was adjusted before use per the manufacturer's instruction. [Y]Exact values unavailable, percentage was estimated based on company product description. [Z]Values reported as lb./ft.3 and g/cm3 to satisfy manufacturers specifications and scientific literature, respectively.

The cut portion of the cutting's internodes were dipped in for three to five seconds. The Hx Vitamin B1 and Hormone (Hormex, Westlake Village, CA) liquid concentrate was diluted to a 1:5 ratio with distilled water, reducing the packaged 2,400 ppm NAA to the desired 500 ppm NAA, before propagules were soaked for a duration of 5 minutes.

## 2.3 Hemp cultivars and cutting preparation

Eight "high-CBD, low-THC" hemp cultivars to include 'ACDC' (A), 'Cherry' (C), and 'Super CBD' (D) (Ano Colorado, LLC; Hartsel, CO), 'ACDC' (M) and 'JL Baux' (L) (Colorado Hemp Institute, LLC.; Parachute, CO), and 'Cherry Wine' (N), 'Mother Earth' (Q), and 'Wife' (P) (Lone Star Valley, LLC.; Monte Vista, CO) were evaluated. The cultivars selected have reported THC $< 0.3\%$ and CBD $\geq 7\%$ which equates to a CBD:THC ratio of $\geq 20:1$. Mother stock plants (5 plants of each cultivar cloned from a singular mother plant) were cultivated in a greenhouse located in Apopka, Florida, United States (latitude 28˚38' N, longitude 81˚33' W) using 1000 W, 7500K metal halide lamps to maintain a photoperiod of 18 hr daylight to ensure plants remained vegetative. Mother stock were maintained in c1600 (13.55 L) containers with Pro-Mix HP Mycor-rhizae (Premier Tech Horticulture, Quakertown, PA, U.S.) media. Mother stock was supplied with Osmocote 15-9-12 5-6-month slow-release fertilizer (Everris NA, Inc.; Dublin, OH, United States) as needed. Mother stock plants were roughly 7 months old when cuttings were taken (11/7/2019).

A total of 120 cuttings of each cultivar were collected from 5 genetically identical mother stock plants. Cuttings were excised from apical shoot tips at a length of ~15 cm. To maintain uniformity, clones were selected with four identifiable nodes and expanded fan leaves. Lower fan leaves were removed and the leaf at the youngest/ apical node were tipped. A 45-degree cut was made below (~1 cm) the fourth node to increase vascular tissue surface area exposure. Although studies have identified that leaf removal and tipping may reduce rooting success of *Cannabis* [13], under our lower humidity rooting conditions excessive leaf tissue resulted in low turgor presser of clones within all propagation medias. Cuttings were rooted in 6.35 cm round Hydrofarm CK64002 plastic 72 cell pack (Hydrofarm, Medley, FL, U.S.) with experimental rooting mediums.

## 2.4 Experimental design and environmental rooting conditions

The experiment was designed as a $4 \times 5 \times 8$ confounded split-split-plot, with the main plot of propagation media (Grodan Rockwool, PRO-MIX HP, Berger BM6, and Jolly Gardner Pro-Line C/20), the sub-plot of rooting hormone application (Hx, DNG, H2, H1, and control), and the sub-sub-plot of eight hemp cultivars: 'ACDC' (A), 'Super CBD' (B), 'Cherry' (C), 'JL Baux' (L), 'ACDC' (M), 'Cherry Wine' (N), 'Wife' (P), and 'Mother Earth' (Q). Each of the 160 treatment combinations was replicated six times, once per tray, for a total of 960 experimental observations.

Clones were placed in environmentally controlled propagation rooms. Room temperature was controlled by an air conditioner set to 25˚C. Air temperature ($24.90 \pm 0.04˚$ C) and relative humidity ($63.62 \pm 0.16\%$) were recorded by thermocouples near canopy height and were collected by a wireless data logging station (HOBO RX3000; Onset Computer Corporation, Bourne, MA, United States). A 24-h photoperiod was provided by T5 fluorescent lamps (E-conolight, Sturtevant, WI, United States). The photosynthetic photon flux density (PPFD) on the propagation bench was measured by a quantum sensor (MQ-500; Apogee Instruments Inc., Logan, UT, United States) at ten representative positions at the seedling canopy level. The average PPFD that cuttings and seedlings received was 54 μmol m$^{-2}$ s$^{-1}$, with a daily light integral (DLI) of 5 mol m$^{-2}$ d$^{-1}$. Media was manually saturated as necessary to maintain cutting turgor.

## 2.5 Data collection

Propagules were harvested for analysis 12 days after sticking (DAS) when more than 50% of all propagules had visible root protrusion from the stem when gently removed from the media [13]. Medium was rinsed from rooted cuttings prior to measurement collection using either water alone or in combination with forceps. Propagules were placed onto a 20" box fan (Lasko; West Chester, PA) and cellulose filter (3M; Saint Paul, MN) for approximately 5 min to remove surface moisture. Roots were then removed from each propagule with a razor blade and recorded for total root number, laid end to end and measured for length, and weighed to record total root mass (**S1 File**). Cuttings that died during the experiment were given a score of zero for the rooting measurements.

## 2.6 Statistical analysis

Statistical analysis was conducted using JMP (JMP®, Version 14. SAS Institute Inc., Cary, NC, 1989–2021.). To normalize data, due to a substantial proportion of zero values within the distribution, the root number, root length, and root mass response variables were transformed using the cube root transformation.

The three-factor split-split plot model (Eq 1) was fitted as follows: where µ is the grand mean;

$$y'_{ijklm} = \mu + T_m + TR_{lm} + \alpha_i + \alpha TR_{ilm} + \beta_j + \alpha\beta_{ij} + \alpha\beta TR_{ijlm} + \gamma_k + \alpha\gamma_{ik} + \beta\gamma_{jk} + \alpha\beta\gamma_{ijk}$$
$$+ \varepsilon_{ijklm} \tag{1}$$

$T_m$ is the random trial effect; $TR_{lm}$ is the random trial-by-rep interaction;, $\alpha_i$ is the fixed, main plot (medium) effect; $\alpha TR_{ilm}$ is the random, error term of the main plot (medium); $\beta_j$ is the fixed, sub-plot effect (hormone); $\alpha\beta_{ij}$ is the fixed, medium-by-hormone interaction effect; $\alpha\beta TR_{ijlm}$ is the random, error term of the sub-plot (hormone) and medium-by-hormone interaction; $\gamma_k$ is the fixed, sub-sub-plot (cultivar) effect; $\alpha\gamma_{ik}$ is the fixed, medium-by-cultivar interaction effect; $\beta\gamma_{jk}$ is the fixed, hormone-by-cultivar interaction effect; $\alpha\beta\gamma_{ijk}$ is the fixed, medium-by-hormone-by-cultivar interaction effect; and $\varepsilon_{ijklm}$ is the residual random error term for sub-sub-plot (cultivar), medium-by-cultivar, hormone-by-cultivar, and medium-by-hormone-by-cultivar. Significant, statistical differences were calculated using Tukey's HSD test ($\alpha < 0.05$).

# 3. Results

Within the three experimental independent variables of medium, hormone, and cultivar and the three recorded root phenotypes of count, length and mass, there were significant ($\alpha = 0.05$) differences observed at $P < 0.001$. All three phenotypes for the medium-by-cultivar interaction effect exhibited significance at $P < 0.001$, whereas only two of the three recorded factors exhibited significant differences for the hormone-by-cultivar interaction effect ($P = 0.019$, $P = >0.05$ (NS), $P = 0.014$). The interaction effects of medium-by-hormone ($P = 0.33$, $P = 0.398$, and $P = 0.531$) and medium-by-hormone-by-cultivar ($P = 0.23$, $P = 0.223$, and $P = 0.299$) lacked significance across all three rooting phenotypes and therefore were omitted.

## 3.1 Rooting media significantly influenced rooting success

Rooting medium significantly affected rooting success with medias ranging from 13.3 to 80.4%. Average rooting success among media treatments expressed 2.26±3.58 roots with a cumulative root length of 3.13±4.24 cm and root mass of 7.39±10.78 g. Cuttings propagated in

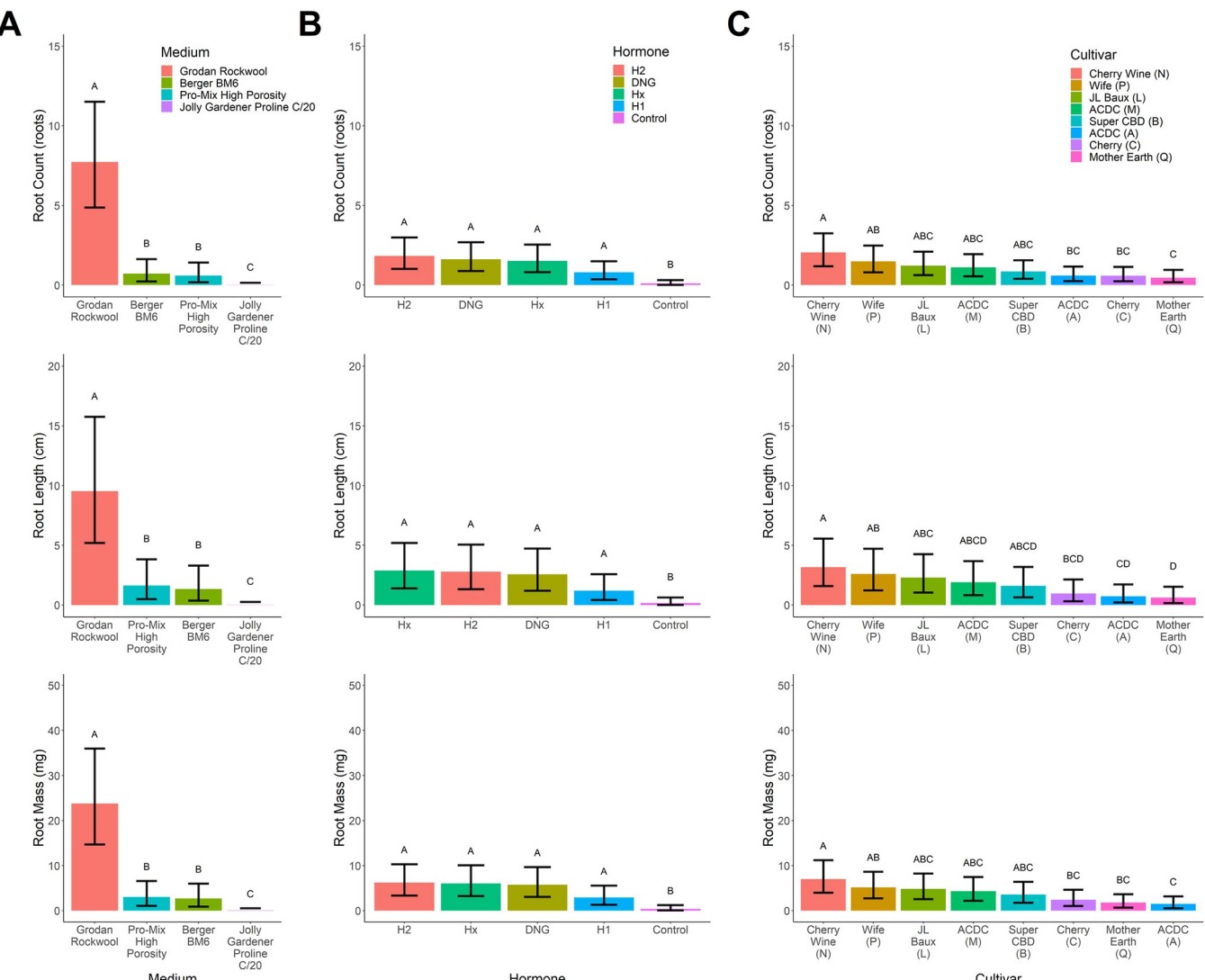

**Fig 2.** Root number (roots; top row), length (cm, center row), and mass (mg; bottom row) for the [**A**] medium, [**B**] hormone, and [**C**] cultivar independent variables for industrial hemp (*Cannabis sativa* L.) vegetative cuttings. Hormone treatments include Hormidin 1 (H1, 1,000 ppm IBA); Hormidin 2 (H2, 3,000 ppm IBA); 1:10 Dip 'N Grow (DNG, 1,000 ppm IBA/500 ppm NAA); 1:5 Hormex (Hx, 500 ppm NAA) and an untreated Control. Bars plot represent group means and error bars represent the 95% confidence interval of each group. Connect letter represent Tukey HSD significant difference tests (α = 0.05).

rockwool displayed the greatest rooting success for root number (7.7 roots), length (9.5 cm), and mass (23.8 mg). Rooting success of rockwool was 7- to 13-fold greater than the next best performing medium for all rooting phenotypes. Rooting success in Berger BM6 (0.7 roots, 1.3 cm, and 2.7 mg) and PRO-MIX HP (0.59 roots, 1.6 cm, and 3.0 mg) were statistically lower than rockwool but not statistically different from each another. Rooting success was lowest in Pro-Line C/20 soilless medium (< 0.1 roots, <0.1 cm, and <0.1 mg) which equated to 435- to 580-fold reduction in rooting success compared to rockwool's performance (**Fig 2A**).

## 3.2 Rooting significantly improved with hormone application

Rooting hormone application also significantly improved success, ranging from 24.0 to 50.5% within hormone treatments. Average rooting success across media treatments (including

control) expressed 1.17±0.63 roots with a cumulative root length of 1.93±0.53 cm and root mass of 4.27±2.23 g. A 2-fold increase in rooting success was observed when hormone was applied compared to no hormone application. No significant differences were observed among hormone treatments (excluding control; 1.44±0.44 roots; 2.36±0.77 cm; 5.24±1.52 g). H2 numerically outperformed other hormone treatments in rooting number and root mass, while Hx resulted in the longest roots. H1 had the least impact on rooting among the hormone treatments (0.8 roots, 1.2 cm, and 2.9 mg) (**Fig 2B**).

### 3.3 Genetic variation in rooting response across cultivars

Significant variance in rooting response was observed across cultivars. Rooting success within cultivars ranged from 37.5 to 46.7%. On average, rooting among cultivars equated to 1.04±0.37 roots with a cumulative root length of 1.73±0.64 cm and root mass of 3.82±1.30 g. 'Cherry Wine' (N) developed the highest root number (2.0 roots), length (3.2 cm), and mass (7.0 mg) among the eight tested "high-CBD" hemp cultivars. 'Cherry Wine' (N) root growth was approximately 20% greater than the next highest rooting cultivar, 'Wife' (P), for all rooting phenotypes. 'Mother Earth' (Q) (0.4 roots, 0.6 cm, and 1.5 mg) had the worst propagule rooting performance among cultivars, expressing a 3.30- to 4.64-fold reduction among rooting phenotypes. (**Fig 2C**).

### 3.4 Cultivar specificity to rooting media or hormone

Significant interaction existed between cultivar and medium, indicating optimal rooting media can vary across essential oil hemp cultivars. Five cultivars (A, L, N, P, and Q) expressed significantly increased rooting response when propagated within rockwool media. 'Super CBD' (B) was unique among cultivars where significantly increased rooting response occurred when propagated within Berger BM6 media (**Fig 3A**). The remaining cultivars (C and M) expressed insignificant rooting responses between PRO-MIX HP, Berger BM6, and rockwool media. All cultivars expressed the least rooting response when propagated with the Proline media and the greatest response in rockwool [excluding 'Super CBD' (B)]. Our results demonstrate cultivar specificity to rooting media exists and can significantly improve rooting success.

Significant interaction existed between cultivar and hormone application, indicating optimal hormone application method and auxin concentrations can vary across essential oil hemp cultivars (**Fig 3B**). For all cultivars, application of auxin versus no auxin applied resulted in improved rooting success. 'Cherry' (C) expressed improved rooting response when liquid, quick-dip applications were utilized. Additionally, 'Cherry' (C) benefited from increased IBA concentration (3000 pmm vs 1000 ppm) if powder application methods are the only products available. Furthermore, 'Super CBD' (B) expressed improved rooting response when propagated with higher concentrations of hormone (1000 ppm IBA + 500 ppm NAA or 3000 ppm IBA). Six cultivars (A, L, M, N, P, and Q) demonstrated low specificity to hormone application method or auxin concentration, which indicates these cultivars are less difficult to root and applicable to a multi-cultivar propagation system.

## 4. Discussion

Being naturally dioecious, hemp is wind pollinated, highly heterozygous, suffers from inbreeding depression, and exhibits hybrid vigor [22]. Furthermore, beyond sexual reproduction, male sex expression is undesirable due to reduced cannabinoid concentrations in essential oil cultivars [23] and reduced fiber quality due to early degradation [6]. Chemical applications can be used to influence sex development of male flowers on female *Cannabis* plants [24], the resulting XX pollen can be used to pollinate female flower to produce "feminized seed" (i.e., all

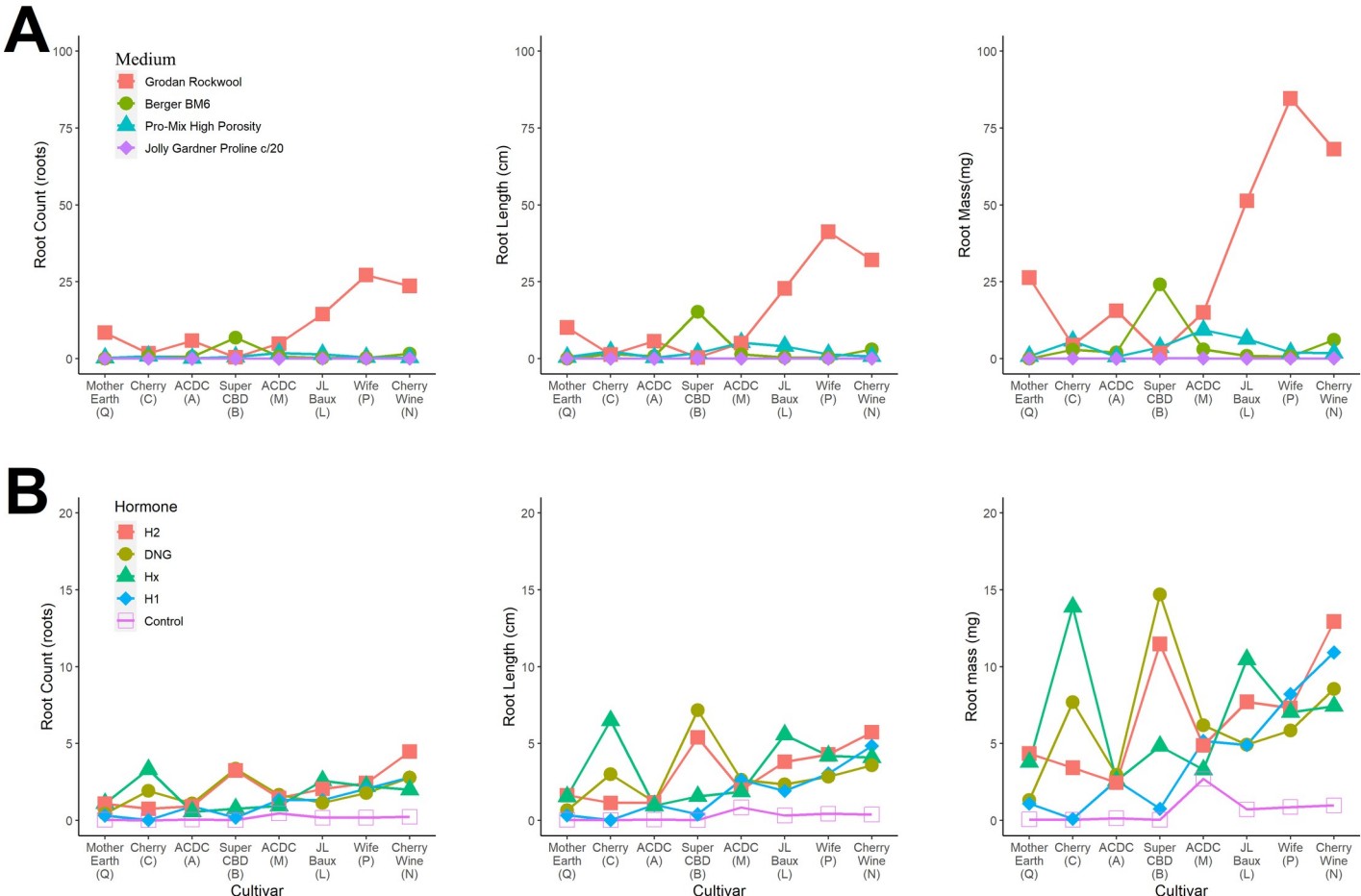

**Fig 3.** Visual representation of root number (roots; left column), length (cm; center column), and mass (mg; right column) for the [**A**] medium-by-cultivar interaction and [**B**] hormone-by-cultivar interaction for industrial hemp (*Cannabis sativa* L.) vegetative cuttings. Hormone treatments include Hormidin 1 (H1, 1,000 ppm IBA); Hormidin 2 (H2, 3,000 ppm IBA); 1:10 Dip 'N Grow (DNG, 1,000 ppm IBA/500 ppm NAA); 1:5 Hormex (Hx, 500 ppm NAA) and an untreated Control. Points are independent and lines are being used for visualization purposes.

resulting seedlings will be XX female genotype) [6, 25, 26]. Unfortunately, propagation through "feminized seed" can demonstrate a high degree of variation with a cultivar [25, 27] at early generation followed by extreme inbreeding after several generations of feminized seed production (i.e., self-pollination) [6]. Mendelian genetics demonstrates that feminized seed must be at the fifth generation of inbreeding ($F_5$) to contain <10% heterozygosity, which should be kept under consideration when looking for stable seed sources [28]. To capitalize on hybrid vigor, feminized $F_1$ hybrid seed should originate from highly inbred parental lineage (>$F_4$) to avoid variation. Alternatively, clonal propagation is used in many crops cultivated by humans (including *Cannabis*) to eliminate variability caused by sexual reproduction, resulting in a consistent performing genotype. Clonal propagation of female plants eliminates the risk of male (XY) genotypes and provides genetically identical plants with the desired phenotypic expression regardless of heterotic levels of the mother plants [6].

### 4.1 Propagation media

Propagation media had greater effect on rooting success compared to hormone and cultivar treatments within our environment. Two media properties which have a significant impact on

rooting success are porosity and bulk density. Highly porous media can be too loose and lead to insufficient contact between the cutting and the media, resulting in loss of turgor [29]. Low porosity can lead to increased water holding capacity of the media and reduced oxygen availability to developing roots [14]. Related to porosity, bulk density (soil weight per unit volume) defines, with higher bulk densities generally having lower total porosity, water holding capacity, and air filled porosity [30]. Selecting a propagation media that offers a balance of good aeration and high water-holding capacity [14, 29, 31] in combination with good water management is critical to rooting success. Of the limited *Cannabis* propagation studies, no comparisons have been made across media types. Campbell et al. [7] used peat-based Grow-Tech Flexi-Plug (Quick Plug, South Portland, ME, USA) with humidity domes and Caplan et al. [13] used PRO-MIX PG Organic growing medium (Premier Tech) with growth chambers under complex environmental conditions. There remains a lack of empirical studies aimed at optimizing *Cannabis* propagation under different production methods (fog, mist, etc.)

When comparing the three commercially available potting mixes utilized in this study, the primary factor considered was the peat concentration, ranging from 65–75%, 80%, and 85% for the PRO-MIX HP, Jolly Gardner Pro-Line C/20, and Berger BM6 medias, respectively. Hemp prefers well-aerated soils with ample aeration and high organic matter concentration; poorly drained or compacted soils can result in difficulty establishing seedlings and young plants. This suggests that rooting success could be improved as aeration is heightened and bulk density is decreased [32]. In the context of bulk density, the greater bulk density reported for the Jolly Garner Pro-Line C/20 versus Berger BM6 and PRO-MIX HP medias could also be responsible for its poor rooting success [33]. The uniformity in density, size, and weight of rockwool carries multiple advantages over commercial potting mixes demonstrated in the study. The uniform medium consistency, optimal air-to-water ratio (due to high porosity and water-holding capacity), and lack of competitive organisms [34–36] resulted in rockwool significantly outperforming the other three medias.

## 4.2 Endogenous auxin application substantially increased rooting

Selection of rooting hormone had minimal impact on rooting success compared to cultivar and medium selection, although, significant improvement was achieved when using a hormone versus the control. Caplan et al. [13] found that the position (apical or basal) *C. sativa* cuttings were taken from had little effect on success or quality of rooting, but that application of a 2,000 ppm IBA treatment exhibited a 2.1-fold increased rate of rooting success and 1.6-fold increased root quality when compared to a 2,000 ppm willow (*Salix alba* L.) extract. Powder based applications of IBA tend to be less effective than solutions of equal concentration (e.g. H1 versus DNG) [20]. Quick-dip solutions have several advantages including application uniformity, reduced cost at large cutting throughput, broad auxin final concentration, and uniform rooting results. Nevertheless, powders are easy to apply, do not require dilutions to achieve desired auxin concentrations, are easy to store, and evidence of application is easily visible [14, 17, 37–39]. *Cannabis* is an easily rooted plant which benefits from the application of auxin. However, excessive auxin concentrations can negatively impact rooting success [17] and such thresholds have not been identified for *Cannabis*.

## 4.3 Management and cultivar selection

Cultivar genetics affected overall rooting success in this study. Comparable results have been reported for marijuana cultivars 'Ghost Train Haze', 'Bubba Kush', and 'Headband' with rooting percentages of 85%, 40%, and 40%, respectively [7]. The differential in rooting identified in

literature and observed in this study is cultivar-specific but can be influenced by environmental and managerial conditions [14]. Proper selection of genetics is a critical consideration for large-scale producers who demand stable genetic rooting response, placing high selection intensity upon rooting success when selecting breeding accessions for mother stock and clonal propagation. If production goals are aimed to provide a diverse portfolio of *Cannabis* genetics, a producer may select to implement a single or limited number of rooting techniques that result in high rooting success across a broad range of cultivars, removing difficult to root cultivars from their system. Adversely, some producers may elect to grow a cultivar that is in high demand in the current market (e.g., CBG cultivar) specializing their management and production environment to optimize rooting success of such cultivars. A comparison could be made between any combination of cultivars with similar market demand and value and should be taken into consideration when incorporating new hemp genetics within a business's supply chain.

## 5. Conclusion

Differing agricultural techniques stemming from cultural preferences interacting with varied environmental factors have led to a diverse industrial hemp phenotype, and the blossoming industrial hemp and cannabinoid markets will demand a reliable supply of consistently cultivated to produce the extracts, consumer goods, and other associated products required. Our study identified: (i) medium selection significantly effects rooting response of hemp vegetative cuttings; (ii) the use of a rooting hormone significantly increases rooting success; (iii) variances in genetics among cultivar effect hemp rooting; and (iv) specific interactions occur between cultivars and medium or hormones, although the substrate or hormone selection may overcome one another. While multiple methods exist for the propagation of industrial hemp, rooting success of vegetative cuttings will be improved by proper selection of genetics and propagation media, in combination with a rooting hormone. Future work could identify additional factors which may influence rooting success including, but not limited to, mother stock nutrient application, age of mother stock, caliper of cutting, length of cutting, number of nodes withing rooting medium, light spectrum/intensity, nutrient applications rates/timing to cuttings, and season/environmental/management rooting plasticity.

## Supporting information

**S1 File. Sheet 1 raw data contains the raw data collection and cube root transformed data used in the statistical analysis.** Sheet 2 Model output means contains the back transformed best linear unbiased predictors/estimates of the model terms.
(XLSX)

## Acknowledgments

The authors would like to thank Jimmie Johnston and Dylan Raab for their assistance with this work. Jerry Fankhauser and Sandra Alomar for administrative assistance; and all members of the University of Florida IFAS Industrial Hemp Pilot Project for their collaboration.

## Author Contributions

**Conceptualization:** Sean M. Campbell, Brian J. Pearson.

**Data curation:** Sean M. Campbell, Steven L. Anderson.

**Formal analysis:** Sean M. Campbell, Steven L. Anderson.

**Funding acquisition:** Zachary T. Brym, Brian J. Pearson.

**Investigation:** Sean M. Campbell, Steven L. Anderson, Brian J. Pearson.

**Methodology:** Sean M. Campbell, Steven L. Anderson, Brian J. Pearson.

**Project administration:** Sean M. Campbell, Steven L. Anderson, Brian J. Pearson.

**Resources:** Steven L. Anderson, Zachary T. Brym, Brian J. Pearson.

**Supervision:** Steven L. Anderson, Zachary T. Brym, Brian J. Pearson.

**Validation:** Sean M. Campbell, Steven L. Anderson.

**Visualization:** Sean M. Campbell, Steven L. Anderson.

**Writing – original draft:** Sean M. Campbell, Steven L. Anderson.

**Writing – review & editing:** Zachary T. Brym, Brian J. Pearson.

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
