## [Decision Letter · Decision Letter 0]

11 Jun 2021

PONE-D-21-08119

Evaluation of substrate composition and exogenous hormone application on vegetative propagule rooting success of essential oil hemp (Cannabis sativa L.)

PLOS ONE

Dear Dr. Anderson II,

Thank you for submitting your manuscript to PLOS ONE. After careful consideration, we feel that it has merit but does not fully meet PLOS ONE’s publication criteria as it currently stands. Therefore, we invite you to submit a revised version of the manuscript that addresses the points raised during the review process.

We look forward to receiving your revised manuscript.

Kind regards,

Bi-Cheng Dong

Academic Editor

PLOS ONE

Journal Requirements:

“The authors would like to thank Mengzi Zhang, Jimmie Johnston, and Dylan Raab for their assistance with this work. The authors would also like to thank Roseville Farms and Green Roads for their financial support to conduct this research and for their support of the UF/IFAS Industrial Hemp Pilot Project. Hemp cultivars were donated by Green Roads, LLC and ANO Colorado, LLC.”

“The authors would also like to thank Roseville Farms and Green Roads for their financial support to conduct this research and for their support of the UF/IFAS Industrial Hemp Pilot Project. Hemp cultivars were donated by Green Roads, LLC and ANO Colorado, LLC.”

Additional Editor Comments (if provided):

Thank you for submitting your manuscript to PLOS ONE for review. Your manuscript has already been reviewed by two referees. Both of them are highly positive about your study and agree that your manuscript to some extent fits the acceptance requirement of PLOS ONE. I do agree with them and thus suggest that you improve the manuscript again, taking the comments of the two referees into account.

Reviewers' comments:

Reviewer's Responses to Questions

**Comments to the Author**

1. Is the manuscript technically sound, and do the data support the conclusions?

Reviewer #1: Yes

Reviewer #2: Yes

2. Has the statistical analysis been performed appropriately and rigorously? 

Reviewer #1: Yes

Reviewer #2: Yes

3. Have the authors made all data underlying the findings in their manuscript fully available?

Reviewer #1: Yes

Reviewer #2: Yes

4. Is the manuscript presented in an intelligible fashion and written in standard English?

Reviewer #1: Yes

Reviewer #2: Yes

5. Review Comments to the Author

Reviewer #1: The authors of the present manuscript have conducted a study to determine the influence of three variables (substrate, hormone application and plant cultivar) on the rooting ability of Cannabis. In my perception the objectives of the work are in line with the chosen experimental design, the results obtained are clearly explained and the conclusions are presented in a correct way. I believe that this is a work that provides valuable information regarding the cultivation of cannabis by clonal propagation. The authors clearly explain the novelty and interest of their work. The conclusions are supported by the results obtained. For all these reasons, I consider that the work is suitable for publication in its present form. I would like to make a few comments that the authors may consider in order to improve their manuscript.

Considering the limitation on THC content required by U.S. legislation for commercial cannabis cultivation, can clonal propagation be beneficial to avoid variability in the content of this secondary metabolite? If so, it may be interesting to discuss this in the section regarding the benefits of clonal propagation.

One of the fundamental aspects of the present work is to improve the vegetative propagation of Cannabis cultivars with high CBD content, which allows preserving the desirable phenotype of the maternal plants. Although I do not consider necessary an explanation of what CBD is, it would be positive to mention what a CBD-rich cultivar is, specifying what CBD content these cultivars may have. While the selection of growing media is justified in the text by its bulk density and water retention capacity, the selection of cultivars seems a bit arbitrary.

It is not clear what is meant by the comment on the lines 145 – 148: “Although studies have identified that leaf removal and tipping may reduce rooting success of Cannabis [13], under our lower humidity rooting conditions excessive leaf tissue resulted in low turgor presser of clones within all propagation medias.” The substrate conditions in the present study is described as moisture saturated. Does this comment refer to a previous experiment? Specify.

I am not an expert in Cannabis cultivation, but a 24h photoperiod seems excessive. Add some source regarding this figure (line 242), and specify if this was maintained throughout the experiment.

How was the moisture saturation in the substrate maintained during the experiment? (Line 165)

Did all cuttings maintain turgor during the experiment, none were counted as dead or non-viable at the end of the experiment? This should be specified.

Minor remarks:

Line 71, has not been -> have not been

Lines 262-264, citation needed.

Fix the link to the URL of the first bibliographic source.

Table 1, reference X in the legend is duplicated.

In the captions of figures 1 and 2, put in italics the name of the species.

Reviewer #2: To find out how rooting hormone application and medium selection impact vegetative propagule rooting success of essential oil hemp is quite interesting and logically sounds good. This study was conducted to quantify the effects of commercially available rooting mediums on rooting success of essential oil hemp and tried to determine the genetic variation in rooting response across essential oil hemp cultivars. The experimental design and writing are well but the logic is not sound good in the introduction section, which need to be improved. However, I have some comments which might be contributed to the improvement of the manuscript

Line 99-102: The proposed objectives are very simple, scientifically sounds not good for good quality paper, and should need to be improved by giving the hypothesis or by raising some question.

Line 100: Why author used only commercial auxin products on rooting success of essential oil hemp cultivars?

Line 253-326: Materials and methods section has written well but I found one problem in the whole discussion section. It seems that author repeated the results with supportive references only in discussion part. It should be improve or rewrite by giving the reasons and avoid from repeating the results. I also suggest giving some limitation or perspective in discussion.

Line 335: If multiple methods exist for the propagation of industrial hemp, then what is application of your research in the field or in the industry?? I also suggest to author add some recommendation for future work in conclusion part.

Please also consider the text answers to the following questions and include it into your manuscript at appropriate place:

What is the new finding in this study?

What is the innovation in the methodology part?

How to use your findings in real-world applications?

6. PLOS authors have the option to publish the peer review history of their article (what does this mean?). If published, this will include your full peer review and any attached files.

Reviewer #1: **Yes: **Rubén Portela Carballeira

Reviewer #2: No

---

## [Author Response · Author response to Decision Letter 0]

28 Jun 2021

Reviewer #1: The authors of the present manuscript have conducted a study to determine the influence of three variables (substrate, hormone application and plant cultivar) on the rooting ability of Cannabis. In my perception the objectives of the work are in line with the chosen experimental design, the results obtained are clearly explained and the conclusions are presented in a correct way. I believe that this is a work that provides valuable information regarding the cultivation of cannabis by clonal propagation. The authors clearly explain the novelty and interest of their work. The conclusions are supported by the results obtained. For all these reasons, I consider that the work is suitable for publication in its present form. I would like to make a few comments that the authors may consider in order to improve their manuscript.

Considering the limitation on THC content required by U.S. legislation for commercial cannabis cultivation, can clonal propagation be beneficial to avoid variability in the content of this secondary metabolite? If so, it may be interesting to discuss this in the section regarding the benefits of clonal propagation.

Thank you for the comment. Although there is no guarantee that any cultivar will be compliant from grow-to-grow. Clonal propagation can ensure that the plants should perform like the previous grow when the breeding history and quality of seed is not well defined. Although the genetic architecture underlying cannabinoid development is in its infancy there have been several regions of the genome that can be selected for cannabinoid concentrations, although this does not guarantee compliant THC levels. Additionally, the scientific understanding of cannabinoid concentrations based on management practices and environmental interactions has just begun to be explored.

The authors feel it would not be appropriate to make such claims in the discussion.

One of the fundamental aspects of the present work is to improve the vegetative propagation of Cannabis cultivars with high CBD content, which allows preserving the desirable phenotype of the maternal plants. Although I do not consider necessary an explanation of what CBD is, it would be positive to mention what a CBD-rich cultivar is, specifying what CBD content these cultivars may have. While the selection of growing media is justified in the text by its bulk density and water retention capacity, the selection of cultivars seems a bit arbitrary.

The selection of cultivars was based on what was provided by the donors and mother plants at the appropriate level to take the necessary cuttings needed for this study. Many of the cultivars presented within this study are commonly available and widely grown low THC, high CBD cultivars. We have provided a statement within the M&M regard what constitutes a high CBD hemp cultivar.

L133: All the cultivars selected have reported THC < 0.3% and CBD ≥ 7 % which equates to a CBD:THC ratio of ≥ 20:1.

It is not clear what is meant by the comment on the lines 145 – 148: “Although studies have identified that leaf removal and tipping may reduce rooting success of Cannabis [13], under our lower humidity rooting conditions excessive leaf tissue resulted in low turgor presser of clones within all propagation medias.” 

Good question. There are very few peer-reviewed publications in reference to cannabis vegetative propagation. One of the original papers presenting vegetative propagation in marijuana cultivars presented the findings the reviewer is inquiring about. The authors chose to put these comments in our materials in methods to acknowledge that our methods are different from those previously published, due to the lower humidity environment we conducted our experiment under.

The substrate conditions in the present study is described as moisture saturated. Does this comment refer to a previous experiment? Specify.

Watering was based on visual appearance of propagules and soil from the previous day. Future research is needed to fine tune what moisture levels in specific medias are optimal for rooting. This is likely cultivar specific.

We adjusted the watering statement withing the M&M.

L166: Media was manually saturated as necessary to maintain cutting turgor.

I am not an expert in Cannabis cultivation, but a 24h photoperiod seems excessive.

Yes, it may seem excessive, but Cannabis does not need a rest period to our knowledge. It is common to push growth under 20 to 24 hrs of light. Since we were rooting our cuttings under low intensity fluorescence light the 24 hr light regime is not uncommon in industry. Although we do acknowledge it is not necessary, it does ensure that none of the cultivar’s transition to reproductive phase.

 Add some source regarding this figure (line 242), and specify if this was maintained throughout the experiment.

The authors are not sure what the reviewers are referring to on line 242.

How was the moisture saturation in the substrate maintained during the experiment? (Line 165)

Substrates were check daily in the morning and flats that were showing signs of wilting or dried substrate were saturated with a hose.

Did all cuttings maintain turgor during the experiment, none were counted as dead or non-viable at the end of the experiment? This should be specified.

Good comment. Plants that died were marked as zeros in the dataset. We have specified this within the materials and methods.

L174: Cuttings that died during the experiment were given a score of zero for the rooting measurements.

Minor remarks:

Line 71, has not been -> have not been

Corrected.

Lines 262-264, citation needed.

Citation added.

Fix the link to the URL of the first bibliographic source.

Corrected.

Table 1, reference X in the legend is duplicated.

Corrected.

In the captions of figures 1 and 2, put in italics the name of the species.

Corrected.

Reviewer #2: To find out how rooting hormone application and medium selection impact vegetative propagule rooting success of essential oil hemp is quite interesting and logically sounds good. This study was conducted to quantify the effects of commercially available rooting mediums on rooting success of essential oil hemp and tried to determine the genetic variation in rooting response across essential oil hemp cultivars. The experimental design and writing are well but the logic is not sound good in the introduction section, which need to be improved. However, I have some comments which might be contributed to the improvement of the manuscript

Line 99-102: The proposed objectives are very simple, scientifically sounds not good for good quality paper, and should need to be improved by giving the hypothesis or by raising some question.

Thank you for the comment. The authors disagree with this comment and believe the objects are in line with and of consistent quality to other papers within PLOS ONE and other equal quality journals.

Line 100: Why author used only commercial auxin products on rooting success of essential oil hemp cultivars?

A major objective for this manuscript was to test readily available auxin products. If there is an auxin product readily available on the market that successfully roots cannabis, producers are not going to go out of there way to buy raw auxin and mix it to their desired concentrations. Furthermore, of the limited studies present in cannabis rooting the only hormone comparisons have been one organic auxin product versus one commercial product, which does not show comparisons between auxin concentrations and application methods which our treatments have done.

Line 253-326: Materials and methods section has written well but I found one problem in the whole discussion section. It seems that author repeated the results with supportive references only in discussion part. It should be improve or rewrite by giving the reasons and avoid from repeating the results. I also suggest giving some limitation or perspective in discussion.

The authors believe the discussion is appropriate and in line with the recommendations of PLOS ONE recommendations (https://plos.org/resource/how-to-write-conclusions/). Key findings are summarized within in the first sentence of each paragraph and supporting discussion from previous research is used to build the discussion of why the results occurred. We have added some future work comments as stated below.

“A successful discussion section puts your findings in context. It should include:

1. Summarize the key findings in clear and concise language.

2. Acknowledge when a hypothesis may be incorrect.

3. Place your study within the context of previous studies.

4. Discuss potential future research.

5. Provide the reader with a “take-away” statement to end the manuscript.”

Line 335: If multiple methods exist for the propagation of industrial hemp, then what is application of your research in the field or in the industry??

As stated below the objective of this manuscript was not to define an innovation. We strived to provide empirical data driven results (which is lacking in all areas of cannabis research) to the commercial and research community as it pertains to cannabis vegetative propagation. hope these findings will give production/research groups the information to reduce or bypass the hurdle of experiments with cloning protocols.

I also suggest to author add some recommendation for future work in conclusion part.

We have included a statement regarding some areas of future work.

L337: Future work is necessary to identify additional factors which may influence rooting success including, but not limited to, mother stock nutrient application, age of mother stock, caliper of cutting, length of cutting, number of nodes withing rooting medium, light spectrum and intensity, nutrient applications to rotting cuttings, and season/environmental/management rooting plasticity.

Please also consider the text answers to the following questions and include it into your manuscript at appropriate place:

What is the new finding in this study?

Within the discussion the clear take away are clearly stated to begin each discussion paragraph followed by context for previous studies.

L272: Propagation media had greater effect on rooting success compared to hormone and cultivar treatments within our environment.

L299: Selection of rooting hormone had minimal impact on rooting success compared to cultivar and medium selection, although, significant improvement was achieved when using a hormone versus the control.

L313: Cultivar genetics affected overall rooting success in this study.

What is the innovation in the methodology part?

The objective of this manuscript was not to define an innovation. We strived to provide empirical data driven results (which is lacking in all areas of cannabis research) to the commercial and research community as it pertains to cannabis vegetative propagation. We hope these findings will give production/research groups the information to reduce or bypass the hurdle of experiments with cloning protocols.

L41: Although the ideal combination was not specifically identified in this study, findings provide insight into how rooting hormone application and medium selection impact vegetative propagule rooting success of essential oil hemp.

How to use your findings in real-world applications?

The authors believe this is clearly stated in the final sentence of the manuscript:

L336: “rooting success of vegetative cuttings will be maximized by proper selection of genetics and propagation media, in combination with a rooting hormone.”

---

## [Editor Report · Decision Letter 1]

5 Jul 2021

Evaluation of substrate composition and exogenous hormone application on vegetative propagule rooting success of essential oil hemp (Cannabis sativa L.)

PONE-D-21-08119R1

Dear Dr. Anderson II,

We’re pleased to inform you that your manuscript has been judged scientifically suitable for publication and will be formally accepted for publication once it meets all outstanding technical requirements.

Kind regards,

Bi-Cheng Dong

Academic Editor

PLOS ONE
---

## [Editor Report · Acceptance letter]

14 Jul 2021

PONE-D-21-08119R1 

Evaluation of substrate composition and exogenous hormone application on vegetative propagule rooting success of essential oil hemp (*Cannabis sativa* L.) 

Dear Dr. Anderson II:

I'm pleased to inform you that your manuscript has been deemed suitable for publication in PLOS ONE. Congratulations! Your manuscript is now with our production department. 

Kind regards, 

on behalf of

Dr. Bi-Cheng Dong 

Academic Editor

PLOS ONE